# Applying simultaneous super-resolution and contrast synthesis to routine clinical magnetic resonance images for automated segmentation of knee joint cartilage: Data from the Osteoarthritis Initiative

**Ales Neubert**[*,1]                                     ALES.NEUBERT@CSIRO.AU
**Pierrick Bourgeat**[*,1]                           PIERRICK.BOURGEAT@CSIRO.AU
**Jason Wood**[1]                                          JASON.WOOD@CSIRO.AU
**Craig Engstrom**[2]                                   C.ENGSTROM@UQ.EDU.AU
**Shekhar S. Chandra**[2]                    SHEKHAR.CHANDRA@UQ.EDU.AU
**Stuart Crozier**[2]                                  STUART@ITEE.UQ.EDU.AU
**Jurgen Fripp**[1]                                     JURGEN.FRIPP@CSIRO.AU

[1] *The Australian e-Health Research Centre, CSIRO Health and Biosecurity, Herston, Australia*
[2] *School of Information Technology and Electrical Engineering, The University of Queensland, St Lucia, Australia*

**Editors:** Under Review for MIDL 2019

## 1. Introduction

High resolution 3D MR images are well suited for automated cartilage segmentation in the human knee joint. However, volumetric scans such as 3D Double-Echo Steady-State (DESS) images, are not routinely acquired. Instead, typical clinical knee MR imaging exams involve acquisition of a series of 2D turbo spin echo (TSE) sequences. TSE images typically have high in-plane resolution (*e.g.* 0.4 mm), but large slice thickness (*e.g.* 3 mm). The cartilage visualization in the individual 2D TSE images is prone to partial volume artifacts due to the thick slices and high cartilage curvature, often resulting in cartilage appearing thinner than it actually is (Figure 1). Consequently, 2D TSE images of the human knee joint are not well suited for automatic cartilage segmentation. In this work, a patch-based UNet convolutional neural network is employed for synthesizing artificial 3D DESS scans (Syn-DESS) from 2D TSE. An automatic segmentation method is then employed to assess the suitability of the Syn-DESS images for knee cartilage segmentation.

## 2. Method

### 2.1. Data

821 examinations (both left and right knees) from 214 subjects from the Osteoarthritis Initiative[1] were used. Sagittally-acquired 3D DESS with water excitation (160 slices, 0.7mm

---

\* Contributed equally

1. https://data-archive.nimh.nih.gov/oai

slice thickness and $0.37 \times 0.37$mm in-plane resolution) was used as the target contrast, sagittal intermediate weighted TSE with fat suppression (37 slices, 3mm slice thickness and $0.36 \times 0.36$mm in-plane resolution) were used as input data. Manual segmentations were available on 88 cases which were used as the testing dataset. The rest was split into training (80%) and validation (20%). The TSE were rigidly aligned to the DESS and resampled using BSpline.

## 2.2. Network

In this work, we use overlapping patches of size $32^3$ voxels with 25% overlap for training, and patches of size $64^3$ voxels with 25% overlap for inference, a scheme similar to that used in recent MR super-resolution work (Chaudhari et al., 2018). The intensity of the entire image was normalized to zero mean and unit variance before patch extraction.

The network was based on the 3D UNet design (Ronneberger et al., 2015). We used 4 encoding levels, doubling the number of channels at each level (starting at 64 for first level). The loss function was the mean square error (MSE). Training was performed with Adam optimisation, with an initial learning rate set of 0.001. Batch size was set to 30. Validation loss was used as an indicator of learning progression. The model was trained for 20 epochs as the validation loss ceased to improve after this point.

In order to improve the synthesis in the cartilage areas, the model was further trained for 10 epochs using only patches identified as containing cartilage. Doing so ensures that more emphasize is given to the bone-cartilage and cartilage-cartilage interfaces. The cartilage voxels were defined using a segmentation mask obtained on the DESS image using previously validated automated segmentation algorithm (Fripp et al., 2007). Since this segmentation is only used for training, it is not required at inference time.

The quality of the synthesis was assessed using the Dice similarity coefficient (DSC) between the manual segmentations and the automatic segmentations computed using the method of Fripp et al. (2007). It uses active shape models (ASM) for the bone segmentation and a local intensity search method for subsequent cartilage delineation.

## 3. Results

Comparisons of the segmentation results using ASM of the TSE and DESS scan, and syn3D-DESS images with and without the cartilage refinement are presented in Table 1 and illustrated in Figure 1.

Cartilage segmentations computed on the TSE had significantly lower ($p < 0.0001$) DSC than those computed on the DESS. Using syn-DESS images improved the cartilage volume segmentation results compared to TSE, with the greatest improvement in the tibia (+0.17, $p < 0.001$), followed by the patella (+0.09, $p < 0.05$) and femur (+0.07, $p < 0.001$)).

The cartilage refinement significantly improved the DSC in all cartilage plates. The improvement was significant in both femoral and patellar cartilages ($p < 0.0001$). In order to check that the improvement in DSC was not due to the extra training, the base model was trained for an extra 10 epochs using all patches. The extra training did not significantly increase the DSC in any of the cartilage plates (results not shown).

## 4. Conclusions

We have presented a method for generating synthetic 3D DESS-like images with high resolution and favourable contrast characteristics from routine clinical, lower resolution 2D TSE images of the knee joint. The synthetic images, generated using simultaneous contrast synthesis and image super-resolution, significantly improved DSC for the all cartilage plates compared to segmenting the 2D TSE scans directly.

Several convolution network based methods have been recently proposed to directly segment the DESS images (Ambellan et al., 2019; Zhou et al., 2018) and could be applied to directly obtain the segmentations from TSE images instead of generating synthetic images. This would however require a large database of manually segmented data. These segmentation would need to be performed on the DESS given its better cartilage visualization, and be co-registered to the TSE, adding an extra level of error, and this would need to be repeated every time a new sequence is being developed. With this approach, only pairs of the source sequence and DESS need to be acquired, and any segmentation method developed to work on DESS can be applied to the resulting synthetic images.

Table 1: Mean DSC for ASM segmentations on the real DESS, real TSE, and syn3D-DESS images compared to gold standard manual segmentations.

| Dataset | Femur | Tibia | Patella |
|---|---|---|---|
| TSE | 0.686 | 0.575 | 0.544 |
| syn3D-DESS | 0.752 | 0.748 | 0.632 |
| syn3D-DESS (Cartilage refinement) | 0.771 | 0.755 | 0.654 |
| DESS | 0.804 | 0.781 | 0.684 |

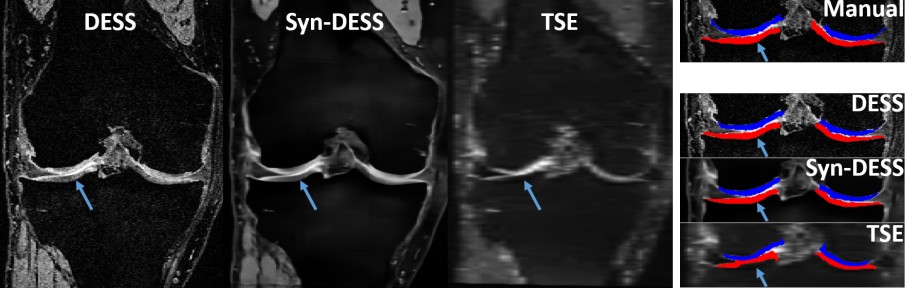

Figure 1: Coronal view on a typical case with DESS, Syn-DESS and BSpline interpolated TSE scan on the left, and manual(top right) and automatic segmentations (bottom right). Each arrow indicates the same location on the bone-cartilage interface.

## Acknowledgments

This research was supported under the Australian Research Councils Linkage Projects funding scheme No. LP100200422 and National Health and Medical Research Councils Development Grant No. APP1091996. The OAI is a public-private partnership comprised of five contracts (N01-AR-2-2258; N01-AR-2-2259; N01-AR-2-2260; N01-AR-2-2261; N01-AR-2-2262) funded by the National Institutes of Health, a branch of the Department of Health and Human Services, and conducted by the OAI Study Investigators. Private funding partners include Merck Research Laboratories; Novartis Pharmaceuticals Corporation, GlaxoSmithKline; and Pfizer, Inc. Private sector funding for the OAI is managed by the Foundation for the National Institutes of Health. This manuscript was prepared using an OAI public use data set and does not necessarily reflect the opinions or views of the OAI investigators, the NIH, or the private funding partners.

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
