# OpenReview forum: "Applying simultaneous super-resolution and contrast synthesis to routine clinical magnetic resonance images for automated segmentation of knee joint cartilage"
_MIDL.io/2019/Conference/Abstract — MIDL Abstract 2019_

### Official Review · AnonReviewer1 · 2019-04-29
**Good performance but motivation for the proposed approach is unclear**

**Rating:** 2
**Confidence:** 2

**Review:**

The paper proposes contrast-synthesis as an intermediate step for their automated segmentation tool, in which they achieve great performance improvement. However, I have a few concerns: firstly, since the network had to “synthesize” cartilage, it is very important to characterize the bias of the network (i.e. does the model consistently over- or underestimate the cartilage size?). Secondly, only the dice score for cartilage segmentation is provided as quantitative results. If one only cares about the cartilage areas, why not just directly train in that area, rather than the refinement approach? Finally, I don’t necessarily agree with their last claim in the conclusion. For this abstract, they already needed to align DESS and TSE images, so “an extra level of error” is already introduced. From this work, they already have sufficiently large dataset of TSE image - DESS segmentation pairs, so it would be interesting to see how the direct segmentation compares to the proposed contrast-synthesis approach.

---

### Official Review · AnonReviewer2 · 2019-04-30
**Convincing results despite limited technical novelty**

**Rating:** 3
**Confidence:** 2

**Review:**

In this paper authors use a patch based U-Net to synthesize artificial 3D DESS scans from a series of 2D TSE images of the knee joint. The paper shows that, when using a traditional active shape model based approach, cartilage segmentation from the artificial 3D DESS is more accurate than directly segmenting 2D TSE images. To further improve the synthesis in cartilage area, after fully training the patch-based U-Net, the training is continued only on patches that contain cartilage, which is demonstrated to further improve results.

The individual building blocks in this work are well-established, and I did not feel like I learned much from reading this extended abstract. However, the work is clearly presented, within scope at MIDL, and results show a convincing benefit of the proposed method. As such, I would argue for acceptance as an extended abstract.

Typos:
 p.2: emphasize -> emphasis

---

### Decision · Program_Chairs · 2019-05-06
**Acceptance Decision**

Accept